**Data Availability Statement:** Due to Research Ethics Committee restrictions, the data set is not

# Suicidal thoughts and behaviour among healthcare workers in England during the COVID-19 pandemic: A longitudinal study

Prianka Padmanathan[1,2]*, Danielle Lamb[3], Hannah Scott[4], Sharon Stevelink[4], Neil Greenberg[4], Matthew Hotopf[4], Richard Morriss[5], Rosalind Raine[3], Anne Marie Rafferty[4], Ira Madan[4], Sarah Dorrington[4], Simon Wessely[4], Paul Moran[1]

**1** Centre for Academic Mental Health, Population Health Sciences, University of Bristol, Bristol, United Kingdom, **2** Avon and Wiltshire Mental Health Partnership NHS Trust, Bristol, United Kingdom, **3** Department of Applied Health Research, University College London, London, United Kingdom, **4** Institute of Psychiatry, Psychology & Neuroscience, King's College London, London, United Kingdom, **5** The Institute of Mental Health, University of Nottingham, Nottingham, United Kingdom

* Prianka.padmanathan@bristol.ac.uk

## Abstract

### Background

During the COVID-19 pandemic, concern has been raised about suicide risk among healthcare workers (HCWs). We investigated the incidence risk and prevalence of suicidal thoughts and behaviour (STB), and their relationship with occupational risk factors, among National Health Service HCWs in England between April 2020 and August 2021.

### Methods

In this longitudinal study, we analysed online survey data completed by 22,501 HCWs from 17 NHS Trusts at baseline (Time 1) and six months (Time 2). The primary outcome measures were suicidal ideation, suicide attempts, and non-suicidal self-injury. We used logistic regression to investigate the relationship between these outcomes and demographic characteristics and occupational factors. Results were stratified by occupational role (clinical/non-clinical).

### Results

Time 1 and Time 2 surveys were completed by 12,514 and 7,160 HCWs, respectively. At baseline, 10.8% (95% CI = 10.1%, 11.6%) of participants reported having experienced suicidal thoughts in the previous two months, whilst 2.1% (95% CI = 1.8%, 2.5%) of participants reported having attempted suicide over the same period. Among HCWs who had not experienced suicidal thoughts at baseline (and who completed the Time 2 survey), 11.3% (95%CI = 10.4%, 12.3%) reported such thoughts six months later. Six months after baseline, 3.9% (95% CI = 3.4%, 4.4%) of HCWs reported attempting suicide for the first time. Exposure to potentially morally injurious events, lack of confidence about raising safety concerns and these concerns being addressed, feeling unsupported by managers, and providing a

publicly available. Requests to access to the de-identified data set can be made to the NHS CHECK team at nhscheck@kcl.ac.uk.

**Funding:** Funding for main NHS CHECK cohort study (co-chief investigators SW, SAMS, RR, MH, NG) was received from the following sources: Medical Research Council (MR/V034405/1); UCL/Wellcome (ISSF3/ H17RCO/C3); Rosetrees (M952); Economic and Social Research Council (ES/V009931/1); NHS England and NHS Improvement; as well as seed funding from National Institute for Health Research Maudsley Biomedical Research Centre, King's College London, National Institute for Health Research Health Protection Research Unit in Emergency Preparedness and Response at King's College London. P.M. is supported by the NIHR Applied Research Collaboration (ARC West) and the NIHR Biomedical Research Centre at University Hospitals Bristol and Weston NHS Foundation Trust and the University of Bristol. The funders had no role in study design, data collection and analysis, decision to publish, or preparation of the manuscript.

**Competing interests:** MH, RR, and SW are senior NIHR Investigators. SW has received speaker fees from Swiss Re for two webinars on the epidemiological impact of COVID 19 pandemic on mental health. SW is a Non Executive Director of NHS-England. RR reports grants from DHSC/UKRI/ESRC COVID-19 Rapid Response Call, grants from Rosetrees Trust, grants from King's Together rapid response call, grants from UCL (Wellcome Trust) rapid response call, during the conduct of the study; & grants from NIHR outside the submitted work MH reports grants from DHSC/UKRI/ESRC COVID-19 Rapid Response Call, grants from Rosetrees Trust, grants from King's Together rapid response call, grants from UCL Partners rapid response call, during the conduct of the study; grants from Innovative Medicines Initiative and EFPIA, RADAR-CNS consortium, grants from MRC, grants from NIHR, outside the submitted work. SS reports grants from UKRI/ESRC/DHSC, grants from UCL, grants from UKRI/MRC/DHSC, grants from Rosetrees Trust, grants from King's Together Fund, and an NIHR Advanced Fellowship [ref: NIHR 300592] during the conduct of the study. NG reports a potential COI with NHSEI, during the conduct of the study; he is the managing director of March on Stress Ltd which has provided training for a number of NHS organisations. NG is not aware if the company has delivered training to any of the participating trusts, as he is not involved in commissioning specific pieces of work. DL is funded by the National Institute for Health and Care Research ARC North

reduced standard of care were all associated with increased suicidal ideation among HCWs during the COVID-19 pandemic. At six months, among clinicians, a lack of confidence about safety concerns being addressed, independently predicted suicidal ideation.

## Conclusion

Suicidal thoughts and behaviour among healthcare workers could be reduced by improving managerial support and enhancing the ability of staff to raise safety concerns.

## Background

There have been longstanding concerns regarding the rate of suicide among healthcare workers (HCWs) [1–3]. The COVID-19 pandemic has placed additional burdens on HCWs, and there is evidence of potentially high rates of adverse mental health outcomes during this time [4–6]. Furthermore, several high-profile cases of suicide among HCWs featured in the media have highlighted the emotional toll of the pandemic on doctors, nurses, and first responders [7–9]. Yet, alarming claims about the impact of the pandemic on suicide in the general population have not accurately reflected suicide statistics and have the potential to cause harm [10, 11]. Empirical evidence about suicidal thoughts and behaviour among HCWs during the pandemic is, therefore, important.

A systematic review of studies reporting on the impact of COVID-19 on suicidal thoughts and behaviour (STB) and self-harm amongst HCWs, which included studies published until May 2021, found the quality of the available literature to be poor [12]. For example, none of the included studies provided follow-up data and many used convenience sampling and did not clearly report their sampling frame. The limited evidence available indicated that certain work settings, such as working with patients with COVID-19 and those with substandard working conditions, were a risk factor for STB and self-harm.

Since the publication of the review by Eyles et al. (2021), several other studies have been published, which investigate the impact of the pandemic on STB among HCWs [13]. These have identified a broader range of potential sociodemographic, clinical and occupational risk factors for STB including a perception of poor working conditions [14] and re-deployment to a COVID-19 setting [15]. However, studies providing prospective follow-up data remain rare [15].

Occupational risk factors have also been linked to STB outside of the COVID-19 pandemic. A recent meta-analysis of job stressors and suicidality, conducted in a heterogeneous population [16], identified that inadequate support from supervisors and colleagues is associated with the occurrence of STB. An association has also been identified between occupational moral injury and suicidality, although until the pandemic, the relationship had mainly been investigated in a small number of studies of military personnel [17]. Among HCWs working across a range of epidemics, several modifiable occupational risk factors, such as a lack of specialist training, or access to personal protective equipment (PPE) and poor availability of effective supervision or support, have been associated with worse mental health [18–20].

This study aimed to improve the current level of understanding of STB among HCWs during the COVID-19 pandemic, using data collected as part of a large-scale repeated online survey of National Health Service (NHS) HCWs. We set out to investigate the incidence risk and prevalence of suicidal ideation, suicide attempts, and non-suicidal self-injury, and the relationship between these thoughts and behaviours and modifiable occupational risk factors, among NHS HCWs during the pandemic.

Thames. PM reports grants from NIHR and the Cassell Hospital Charitable Trust outside the submitted work. PM is part-funded by the NIHR ARC West. PP and PM report a grant from Bristol and Weston Hospitals Charity outside the submitted work. PP was funded by the Medical Research Council Addiction Research Clinical Training programme (MR/N00616X/1). Other authors report no competing interests. The views expressed in this publication are those of the authors and not necessarily those of the National Institute for Health Research or the Department of Health and Social Care. This does not alter our adherence to PLOS ONE policies on sharing data and materials.

## Methods

In this study, we analysed data from a longitudinal online survey (NHS Check) that was distributed to all HCWs (clinical and non-clinical), students, and volunteers in 18 NHS Trusts across England during the COVID-19 pandemic [21]. Data for Time 1 were collected between April 2020 and January 2021. Data for Time 2 were collected six months after each HCW's completion of the Time 1 survey, from October 2020 to August 2021.

### Context

To date, in England, there have been two major peaks in the incidence of COVID-19 deaths (April 2020 and January 2021) [22]. Three national lockdowns have been enforced (March-June 2020; October-December 2020; January-July 2021), although the level of restrictions within each lockdown varied considerably. Between national lockdowns, some regional restrictions were also applied [23]. Data collection in this study commenced a month after the first lockdown and ended a month after the third lockdown.

### Data collection

The NHS Check survey link was shared in emails to all HCWs sent by senior Trust management, promoted in team meetings/briefings, included in newsletters, and advertised on Trust intranets and social media groups and via screen savers on Trust computers. The surveys collected data on socio-demographic characteristics, occupational role and work setting, experiences of COVID-19, and a series of validated measures (detailed below). To improve acceptability and therefore participation rates, the initial Time 1 survey was designed to be very short (completed in 5 minutes), and HCWs could then opt-in to a longer set of Time 1 questions, which included the suicide questions.

### Outcomes

Suicidal ideation, suicide attempts, and non-suicidal self-injury were assessed using the following items from the Clinical Interview Schedule (CIS-R) [24]: *"Have you ever thought of taking your life, even though you would not actually do it?"* (suicidal ideation); *"Have you ever made an attempt to take your life, by taking an overdose of tablets or in some other way?"* (suicide attempts); and *"Have you ever deliberately harmed yourself in any way but not with the intention of killing yourself?"* (non-suicidal self-injury). In the Time 1 survey, HCWs could select from the following options: *"Yes, in the past 2 months"*, *"Yes, but not in the past 2 months"*, or *"No"*. In the Time 2 survey, HCWs could select from the same options but a one, rather than two, month time frame was specified.

### Explanatory factors

We explored demographic and mental health characteristics and individual-level occupational factors. In this study, characteristics were defined as follows based on previous analyses [25]: age, sex (female, male–options were limited by the survey design), ethnicity (top-level Office for National Statistics categories [26]: White, Black, Asian, Mixed ethnicity, other ethnicity), role (clinical, non-clinical; see S1 File for a list of occupational groups), and probable depression (Patient Health Questionnaire score ≥10). Anxiety and post-traumatic stress disorder were not included as possible explanatory factors as these were considered likely to be highly correlated [27, 28].

The following individual-level occupational factors were investigated as binary exposures (see S2 File for original questions): re-deployment status; exposure to potentially morally

injurious events (collected using the Moral Injury Events Scale [29, 30]); lack of access to personal protective equipment (PPE); lack of confidence about raising safety concerns; lack of confidence that safety concerns would be addressed; feeling unsupported by supervisors or managers, and providing a reduced standard of care. These factors were selected a priori from the NHS CHECK survey by co-authors P.P., D.L., and P.M. on the basis that they were theoretically likely to be associated with STB, modifiable, and/or of interest for policy or practice. They were pre-specified in our published protocol [31].

## Statistical analysis

Data from 17 Trusts (only those with a response rate of over 5%) were included in this analysis. The Time 1 full cohort was weighted using a ranking algorithm based on the age, sex, ethnicity, and roles profile of the workforce at each Trust to maximise representativeness. To complete the weighting, missing demographic data were imputed using multiple imputation. The imputed data were only used to complete weighting and were not used in any other analyses in this study. Some participants erroneously completed the baseline survey more than once. For these participants, their most complete response was included, unless duplicate responses were equally complete, in which case the earliest of these responses was included.

We first described the demographic and occupational characteristics of the full cohort, Time 1 subsample, and Time 2 subsample. We then described participant responses to survey questions about occupational factors and probable depression at Time 1. Next, we calculated the prevalence of our outcomes at Time 1 and at Time 2. In the Time 2 survey, there was a typographical error whereby the suicide-related questions asked about these behaviours having occurred over the previous month rather than two months, as had been asked in the Time 1 survey. As such, a two-month period prevalence was calculated for Time 1, whilst a one-month period prevalence was calculated for Time 2. In addition to the prevalence, we calculated the incidence risk of our outcomes at Time 2 i.e., the number and percentage of participants who responded "no" to previous self-harm and suicidal behaviour at Time 1, but "yes" at Time 2.

To investigate the relationship between demographic characteristics and occupational factors at Time 1 and suicide-related outcomes at Time 1 and Time 2, we used two-level random-effects logistic regression models (I.e., multilevel analysis) to account for clustering of the HCW data within each Trust. We stratified data by occupational role (clinical/non-clinical), as pre-specified in our published protocol, due to previous NHS Check analyses identifying differences in mental health outcomes by role [25]. Binary suicide-related outcomes were used in this analysis, where the presence of an outcome represented its occurrence within the specified time period. HCWs (level 1) were clustered within each Trust (level 2), and we adjusted the odds ratios for age, sex, ethnicity, and date of survey completion as level 1 covariates. Due to the prospective nature of the analyses between Time 1 explanatory factors and Time 2 suicide-related outcomes, we also adjusted for the corresponding outcome at Time 1 (level 1). For example, when analysing suicidal ideation at Time 2, we adjusted for suicidal ideation at Time 1. As a post-hoc sensitivity analysis, when analysing the relationship between occupational factors and suicide-related outcomes, we also adjusted a priori for probable depression (based on a PHQ-9 score$\geq$10) collected at Time 1 (S3 File). Additionally, cross-sectional logistic regression models for Time 2 are included in S4 and S5 Files. In each model, we only included participants with complete data on all the variables that were included within the model. All analyses were conducted using Stata v 17.0. The "subpop" command in Stata was used to ensure that the standard errors of the estimates were calculated correctly [32].

Ethics. Ethical approval for the NHS CHECK study was granted by the Health Research Authority (reference: 20/HRA/210, IRAS: 282686) and local Trust Research and Development approval. A protocol pre-specifying the analysis plan has been published [31].

## Results

Fig 1 shows the flow of participants: the Time 1 full cohort (who completed the initial short Time 1 survey, which did not include the suicide questions, and on whom the weighting described in the Methods was based; n = 22,501); a subset of the full cohort who also completed the Time 1 suicide survey questions (termed throughout the rest of the manuscript Time 1 sample; n = 12,514); a subset of the Time 1 sample who also completed the Time 2 suicide survey questions (Time 2 sample; n = 7,160). The overall survey response rate was approximately 16% (22,501 responses in the full sample from a total Trust population of 139,037 employed HCWs).

Across the samples (Table 1), most survey respondents were female (full cohort: 73.2%) and of White ethnicity (full cohort: 75.0%). Approximately one in ten participants in each sample were doctors, the remainder were nurses, or had other clinical or non-clinical backgrounds (approximately a third in each role category). Differences in the socio-demographic profiles of the full cohort, Time 1 sample and Time 2 sample are shown in Table 1.

At Time 1, about one in ten participants had been re-deployed (clinical: 12.9%; non-clinical; 9.5%) (Table 2). A lack of access to PPE was reported by 9.2% of clinical participants and 14.5% non-clinical participants, whilst 20.5% of clinical participants and 17.2% of non-clinical participants reported feeling unsupported by managers. Among clinical participants, 9.5% lacked confidence raising safety concerns, whilst 16.4% lacked confidence that safety concerns would be addressed. Among non-clinical participants, 7.0% lacked confidence raising safety concerns, whilst 9.1% lacked confidence that safety concerns would be addressed. A reduced standard of care provided was reported by 16.5% of clinical participants, and 11.0% of non-

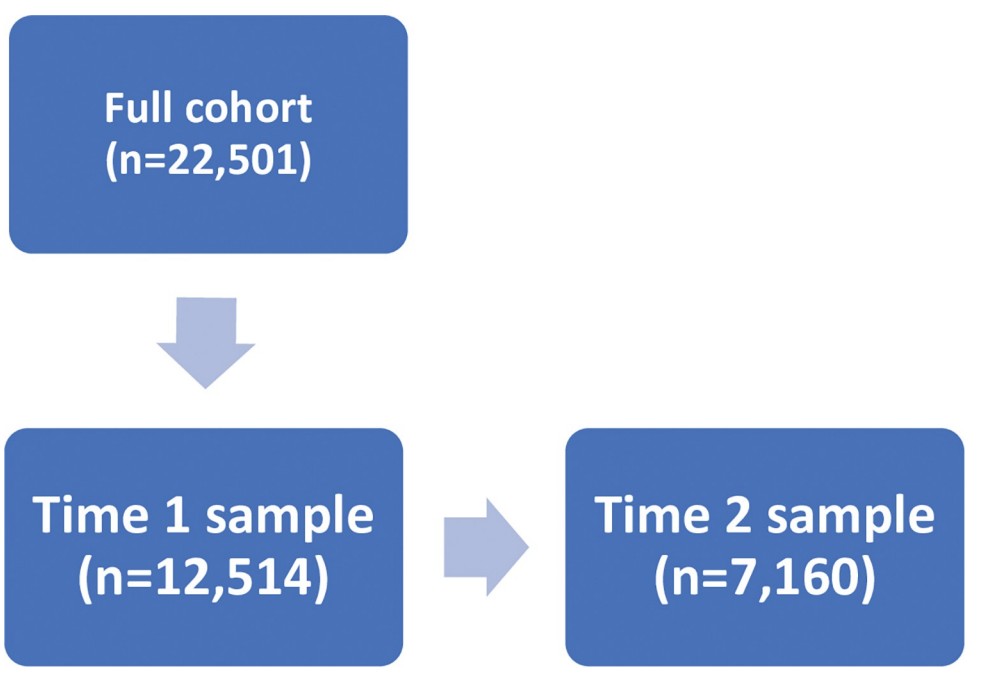

**Fig 1. Flow diagram of samples included within this study.**

**Table 1. Number and proportion of participants by demographic and occupational characteristics in the full cohort, Time 1 sample, and Time 2 sample.**

| Variable | Category | No. (%) unless otherwise specified | | |
|---|---|---|---|---|
| | | Full cohort (n = 22,501) | Time 1 sample (n = 12,514) | Time 2 sample (n = 7,160) |
| Age in years[α] | ≤30 | 4,277 (20.6) | 2,190 (18.7) | 1,075 (15.7) |
| | 31–40 | 4,939 (25.4) | 2,701 (25.4) | 1,425 (23.4) |
| | 41–50 | 5,611 (22.6) | 3,151 (23.0) | 1,869 (24.2) |
| | 51–60 | 5,214 (19.8) | 3,106 (21.3) | 1,962 (24.2) |
| | ≥61 | 1,304 (6.2) | 790 (6.9) | 507 (7.8) |
| | Missing | 1,156 (5.5) | 576 (4.7) | 322 (4.6) |
| Sex[α] | Female | 18,067 (73.2) | 10,342 (75.7) | 5,874 (75.2) |
| | Male | 4,172 (25.5) | 2,110 (23.7) | 1,255 (24.4) |
| | Missing | 262 (1.3) | 62 (0.6) | 31 (0.5) |
| Ethnicity[α] | White | 19,085 (75.0) | 11,159 (80.9) | 6,453 (83.2) |
| | Black | 989 (8.6) | 345 (6.0) | 179 (5.3) |
| | Asian | 1,481 (12.0) | 614 (9.6) | 313 (8.3) |
| | Mixed/Multiple racial & ethnic groups | 546 (1.2) | 278 (1.2) | 154 (1.1) |
| | Other racial & ethnic minority groups | 207 (2.3) | 86 (2.0) | 45 (1.9) |
| | Missing | 193 (1.0) | 32 (0.2) | 16 (0.2) |
| Role[α] | Doctor | 1,653 (10.0) | 835 (9.4) | 479 (9.5) |
| | Nurse | 5,741 (28.5) | 3,366 (29.8) | 1,829 (28.3) |
| | Other clinical | 6,778 (31.1) | 3,646 (30.4) | 1,993 (29.6) |
| | Non-clinical | 7,977 (29.0) | 4,571 (29.8) | 2,802 (32.0) |
| | Missing | 352 (1.5) | 96 (0.6) | 57 (0.6) |

n = frequencies (unweighted raw data), percentages are weighted

clinical participants. About a quarter of participants reported exposure to potentially morally injurious events (clinical: 28.6%; non-clinical 24.3%), whilst a similar proportion met the criteria for "probable depression" based on their PHQ-9 scores (clinical: 27.7%; non-clinical: 29.3%).

## Prevalence of suicidal thoughts and behaviour

Table 3 displays the period prevalence of STB at Time 1 and Time 2. At Time 1, 10.8% (95% CI = 10.1%, 11.6%) of participants reported having experienced suicidal thoughts in the previous two months, whilst 2.1% (95% CI = 1.8%, 2.5%) of participants reported having attempted suicide over the same period. At Time 2, 9.0% (95% CI = 8.1%, 9.9%) of participants reported suicidal ideation and 2.4% (95% CI = 2.0%, 2.9%) reported attempting suicide in the previous month.

## Incidence risk of suicidal thoughts and behaviour

At Time 2, the estimated incidence risk of first-time suicidal ideation was 11.3% (95% CI = 10.4%, 12.3%; n = 524/4,625 people who had not reported previous suicidal ideation at Time 1 and completed the Time 2 survey). The estimated incidence risk of first-time suicide attempts was 3.9% (95% CI = 3.4%, 4.4%; n = 244/6,323 people who had not reported previous suicide attempts at Time 1 and completed Time 2). The estimated incidence risk of first-time non-suicidal self-injury was 6.1% (95% CI = 5.5%, 6.7%; n = 360/5,949 people who had not reported previous non-suicidal self-injury at Time 1 and completed Time 2).

**Table 2. Number and proportion of participants by responses to survey questions on occupational risk factors and probable depression at Time 1, stratified by occupational group.**

| Variable | Category | No. (%) unless otherwise specified | | |
|---|---|---|---|---|
| | | Overall (n = 12,514) | Clinical (n = 7,847) | Non-clinical (n = 4,571) |
| Redeployment | No (Ref) | 11,009 (88.2) | 6,822 (87.1) | 4,102 (90.5) |
| | Yes | 1,472 (11.9) | 1,009 (12.9) | 455 (9.5) |
| PPE access | Access (Ref) | 9,752 (89.3) | 6,628 (90.8) | 3,051 (85.5) |
| | Lack of access | 967 (10.7) | 585 (9.2) | 376 (14.5) |
| Managerial support | Supported (Ref) | 10,272 (80.6) | 6,397 (79.5) | 3794 (82.8) |
| | Unsupported | 2,192 (19.5) | 1,428 (20.5) | 752 (17.2) |
| Raising safety concerns | Confident (Ref) | 10,536 (91.3) | 6,718 (90.6) | 3,745 (93.0) |
| | Lack of confidence | 958 (8.8) | 691 (9.5) | 260 (7.0) |
| Safety concerns being addressed | Confident (Ref) | 9,995 (85.7) | 6,252 (83.6) | 3,675 (91.0) |
| | Lack of confidence | 1,491 (14.3) | 1,148 (16.4) | 331 (9.1) |
| Standard of care provided | Not reduced (Ref) | 8,623 (84.6) | 6,150 (83.5) | 2,414 (89.0) |
| | Reduced | 1,640 (15.4) | 1,352 (16.5) | 281 (11.0) |
| Potentially morally injurious events | No exposure (Ref) | 8,804 (72.7) | 5,369 (71.4) | 3,366 (75.7) |
| | Exposure | 2,898 (27.3) | 1,951 (28.6) | 931 (24.3) |
| Probable depression (PHQ-9 score ≥10) | No (Ref) | 8,650 (71.8) | 5,458 (72.3) | 3,127 (70.7) |
| | Yes | 3,259 (28.2) | 1,999 (27.7) | 1,237 (29.3) |

n = frequencies (unweighted raw data), percentages are weighted

Where numbers within a variable do not equal the total number within the sample, the remainder of the responses were missing. The numbers for missing data are provided in S6 File.

## Factors associated with suicidal thoughts and behaviour

Table 4 displays the adjusted odds ratios describing the relationship between demographic characteristics and STB at Time 1 (cross-sectional analyses) and Time 2 (prospective analyses). At Time 1 participants above 30 years were consistently at increased risk of suicidal ideation in both clinical and non-clinical groups compared with participants 30 years of age or below. Where there was statistical evidence of an association between other demographic characteristics and STB over the two time periods, the effects were inconsistent. In the cross-sectional analyses at Time 2 (S4 File), which did not adjust for the corresponding outcome at Time 1, almost all age categories above 30 years were associated with increased suicidal ideation compared with participants aged 30 years or less.

**Table 3. Prevalence of suicidal ideation, suicide attempts, and non-suicidal self-injury among participants at Time 1 and Time 2.**

| Time | Response | Suicidal ideation | | Suicidal attempts | | Non-suicidal self-injury | |
|---|---|---|---|---|---|---|---|
| | | n | % (95% CI) | n | % (95% CI) | n | % (95% CI) |
| Time 1* | No | 8,137 | 65.7 (64.6, 66.7) | 10,927 | 87.2 (86.4, 88.0) | 10,262 | 82.3 (81.4, 83.1) |
| | Yes, but not in previous 2 months | 2,596 | 19.5 (18.7, 20.4) | 880 | 6.7 (6.1, 7.3) | 1,397 | 10.3 (9.7, 11.0) |
| | Yes, within the previous 2 months | 1,336 | 10.8 (10.1, 11.6) | 262 | 2.1 (1.8, 2.5) | 407 | 3.4 (3.0. 3.8) |
| Time 2* | No | 4,308 | 61.4 (60.0, 62.8) | 5,897 | 82.7 (81.5, 83.8) | 5,532 | 78.2 (77.0, 79.4) |
| | Yes, but not in the previous month | 1,591 | 21.0 (19.8, 22.2) | 475 | 6.3 (5.6, 7.0) | 776 | 9.9 (9.1, 10.8) |
| | Yes, within the previous month | 638 | 9.0 (8.1, 9.9) | 164 | 2.4 (2.0, 2.9) | 226 | 3.2 (2.7, 3.7) |

n = frequencies (unweighted raw data); percentages are weighted

Where numbers within a variable do not equal the total number within the sample, the remainder of responses were missing.

*Time 1 survey = 2-month period prevalence; Time 2 survey = 1-month period prevalence

**Table 4. Association between participant demographic characteristics and suicidal ideation, suicide attempts, and non-suicidal self-injury at Time 1 (cross-sectional analyses) and Time 2 (prospective analyses), stratified by occupational role.**

| Demographic characteristics | Category | Time 1 | | | | | | Time 2 | | | | | |
|---|---|---|---|---|---|---|---|---|---|---|---|---|---|
| | | Suicidal ideation (aOR; 95% CI) | | Suicide attempts (aOR; 95% CI) | | Non-suicidal self-injury (aOR; 95% CI) | | Suicidal ideation (aOR; 95% CI) | | Suicide attempts (aOR; 95% CI) | | Non-suicidal self-injury (aOR; 95% CI) | |
| | | Clinical | Non-clinical | Clinical | Non-clinical | Clinical | Non-clinical | Clinical | Non-clinical | Clinical | Non-clinical | Clinical | Non-clinical |
| **Age in years** | ≤30 (Ref) | 1.00 | 1.00 | 1.00 | 1.00 | 1.00 | 1.00 | 1.00 | 1.00 | 1.00 | 1.00 | 1.00 | 1.00 |
| | **31–40** | **0.74 (0.55, 0.99)** | **0.54 (0.36, 0.81)** | 0.58 (0.29, 1.16) | 1.40 (0.56, 3.50) | 0.65 (0.40, 1.05) | 0.88 (0.48, 1.64) | **0.62 (0.45, 0.85)** | 0.51 (0.27, 0.97) | 0.75 (0.30, 1.85) | 0.51 (0.19, 1.37) | 1.01 (0.56, 1.80) | **0.31 (0.13, 0.73)** |
| | **41–50** | **0.51 (0.35, 0.76)** | **0.55 (0.38, 0.81)** | 0.58 (0.29, 1.15) | 1.19 (0.46, 3.06) | **0.55 (0.39, 0.78)** | 0.55 (0.29, 1.05) | 0.78 (0.54, 1.12) | 0.51 (0.33, 0.79) | 1.29 (0.52, 3.18) | 0.51 (0.22, 1.22) | 1.05 (0.51, 2.14) | 0.48 (0.29, 0.79) |
| | **51–60** | **0.48 (0.36, 0.63)** | **0.31 (0.26, 0.37)** | 0.87 (0.36, 2.07) | 0.70 (0.34, 1.42) | 0.54 (0.27, 1.06) | **0.39 (0.22, 0.68)** | **0.51 (0.33, 0.79)** | 0.40 (0.29, 0.56) | 1.05 (0.49, 2.26) | 0.52 (0.25, 1.10) | 0.62 (0.31, 1.24) | 0.42 (0.22, 0.79) |
| | **≥61** | **0.43 (0.23, 0.82)** | **0.24 (0.13, 0.47)** | 1.07 (0.46, 2.50) | 1.45 (0.71, 2.96) | 0.64 (0.30, 1.35) | **0.40 (0.19, 0.85)** | 0.81 (0.45, 1.44) | 0.53 (0.30, 0.94) | 1.65 (0.60, 4.51) | 0.89 (0.30, 2.61) | 1.04 (0.39, 2.77) | 0.73 (0.36, 1.48) |
| **Sex** | Female (Ref) | 1.00 | 1.00 | 1.00 | 1.00 | 1.00 | 1.00 | 1.00 | 1.00 | 1.00 | 1.00 | 1.00 | 1.00 |
| | **Male** | 1.42 (0.93, 2.15) | **1.49 (1.16, 1.93)** | 1.32 (0.74, 2.34) | 1.49 (0.68, 3.24) | 1.04 (0.60, 1.83) | 1.58 (0.99, 2.52) | 0.94 (0.59, 1.48) | 1.00 (0.62, 1.61) | 1.62 (0.92, 2.85) | 0.57 (0.15, 2.11) | 1.05 (0.59, 1.85) | 0.63 (0.24, 1.65) |
| **Ethnicity** | White (Ref) | 1.00 | 1.00 | 1.00 | 1.00 | 1.00 | 1.00 | 1.00 | 1.00 | 1.00 | 1.00 | 1.00 | 1.00 |
| | **Black/African/ Caribbean/ Black British** | 0.64 (0.34, 1.21) | 0.77 (0.44, 1.35) | 2.21 (0.69, 7.06) | 1.42 (0.48, 4.17) | 1.25 (0.38, 4.06) | 1.24 (0.38, 4.08) | 0.49 (0.10, 2.31) | 0.93 (0.22, 3.96) | 1.06 (0.30, 3.72) | 0.68 (0.06, 7.34) | 0.84 (0.26, 2.65) | 0.55 (0.05, 5.60) |
| | **Asian/Asian British** | **0.64 (0.45, 0.91)** | 0.69 (0.31, 1.54) | 1.16 (0.57, 2.34) | 1.25 (0.61, 2.58) | 0.77 (0.37, 1.60) | 1.00 (0.45, 2.20) | 1.30 (0.74, 2.29) | 0.56 (0.32, 0.99) | 2.05 (0.71, 5.89) | 1.50 (0.70, 3.22) | 1.21 (0.48, 3.08) | 1.03 (0.35, 2.99) |
| | **Mixed/ Multiple racial and ethnic groups** | 1.58 (0.90, 2.77) | 0.85 (0.48, 1.50) | 1.55 (0.49, 4.86) | 0.13 (0.01, 1.43) | **3.44 (1.10, 10.73)** | 0.17 (0.01, 2.15) | 0.98 (0.25, 3.87) | 0.80 (0.11, 5.65) | 2.62 (0.84, 8.22) | 0.10 (0.01, 1.11) | 2.22 (0.88, 5.55) | 1.20 (0.24, 6.09) |
| | **Other racial and ethnic minority groups** | 0.80 (0.30, 2.11) | 1.55 (0.33, 7.25) | - | - | 0.86 (0.10, 7.39) | 0.82 (0.08, 7.88) | - | 0.23 (0.02, 2.90) | - | - | - | - |

Statistically significant results are in bold; Empty cell indicate that the sample size was too small to calculate an odds ratio

aOR: adjusted odds ratios: adjusted for age, sex, ethnicity, and date of survey completion (Time 2 models also adjusted for the corresponding outcome measured at Time 1); CI: confidence intervals

Table 5 displays associations between occupational factors at Time 1 and STB at Time 1 (cross-sectional analyses) and Time 2 (prospective analyses). At Time 1, with the exception of redeployment, all occupational factors were associated with increased suicidal ideation in at least one of the clinical or non-clinical groups. At Time 2, there was evidence of an association between lack of confidence about safety concerns being addressed in the clinical group but not the non-clinical group. For some occupational factors, such as lack of access to PPE in the non-clinical groups, the confidence intervals indicated the possibility of a large increase in risk of suicidal ideation, attempts, and non-suicidal self-injury in the non-clinical group. In a post-hoc sensitivity analysis (S3 File), these models were also adjusted for probable depression

**Table 5. Association between occupational factors at Time 1 and suicidal ideation, suicide attempts, and non-suicidal self-injury at Time 1 (cross-sectional analyses) and Time 2 (prospective analyses), stratified by occupational role.**

| Occupational factors at Time 1 | Category | Time 1 | | | | | | Time 2 | | | | | |
|---|---|---|---|---|---|---|---|---|---|---|---|---|---|
| | | Suicidal ideation (aOR; 95% CI) | | Suicide attempts (aOR; 95% CI) | | Non-suicidal self-injury (aOR; 95% CI) | | Suicidal ideation (aOR; 95% CI) | | Suicide attempts (aOR; 95% CI) | | Non-suicidal self-injury (aOR; 95% CI) | |
| | | Clinical | Non-clinical | Clinical | Non-clinical | Clinical | Non-clinical | Clinical | Non-clinical | Clinical | Non-clinical | Clinical | Non-clinical |
| **Redeployment** | No (Ref) | 1.00 | 1.00 | 1.00 | 1.00 | 1.00 | 1.00 | 1.00 | 1.00 | 1.00 | 1.00 | 1.00 | 1.00 |
| | Yes | 1.14 (0.81, 1.62) | 0.98 (0.60, 1.61) | 1.45 (0.75, 2.82) | 1.09 (0.31, 3.80) | 1.34 (0.95, 1.89) | 1.35 (0.53, 3.40) | 1.08 (0.68, 1.72) | 1.42 (0.95, 2.12) | 0.71 (0.28, 1.82) | 1.87 (0.81, 4.32) | 0.83 (0.51, 1.35) | 1.32 (0.68, 2.57) |
| **Raising safety concerns** | Confident (Ref) | 1.00 | 1.00 | 1.00 | 1.00 | 1.00 | 1.00 | 1.00 | 1.00 | 1.00 | 1.00 | 1.00 | 1.00 |
| | Lack of confidence | **2.20 (1.61, 3.01)** | **1.84 (1.11, 3.04)** | 0.84 (0.40, 1.76) | 1.29 (0.58, 2.84) | 1.01 (0.58, 1.74) | **2.44 (1.19, 5.02)** | 1.47 (0.93, 2.31) | 1.72 (0.65, 4.57) | 0.53 (0.20, 1.43) | 0.53 (0.08, 3.36) | 1.07 (0.45, 2.52) | 0.99 (0.38, 2.55) |
| **Safety concerns being addressed** | Confidence (Ref) | 1.00 | 1.00 | 1.00 | 1.00 | 1.00 | 1.00 | 1.00 | 1.00 | 1.00 | 1.00 | 1.00 | 1.00 |
| | Lack of confidence | **2.19 (1.62, 2.95)** | **1.82 (1.31, 2.54)** | 0.91 (0.55, 1.49) | 0.66 (0.28, 1.53) | 1.31 (0.91, 1.89) | 1.24 (0.42, 3.72) | **1.45 (1.12, 1.89)** | 1.31 (0.65, 2.64) | 0.98 (0.51, 1.89) | 2.19 (0.76, 6.32) | 1.47 (0.83, 2.61) | 2.36 (0.88, 6.32) |
| **PPE access** | Access (Ref) | 1.00 | 1.00 | 1.00 | 1.00 | 1.00 | 1.00 | 1.00 | 1.00 | 1.00 | 1.00 | 1.00 | 1.00 |
| | Lack of access | 1.36 (0.91, 2.04) | **1.34 (1.04, 1.73)** | 0.40 (0.10, 1.61) | 1.35 (0.65, 2.77) | 0.86 (0.28, 2.65) | 1.66 (0.71, 3.90) | 1.50 (0.96, 2.32) | 1.57 (0.83, 2.97) | 0.49 (0.11, 2.23) | 2.64 (0.90, 7.75) | 0.65 (0.20, 2.09) | 2.50 (0.99, 6.33) |
| **Managerial support** | Supported (Ref) | 1.00 | 1.00 | 1.00 | 1.00 | 1.00 | 1.00 | 1.00 | 1.00 | 1.00 | 1.00 | 1.00 | 1.00 |
| | Unsupported | **2.18 (1.67, 2.85)** | **1.57 (1.05, 2.34)** | 0.98 (0.61, 1.58) | 0.75 (0.42, 1.32) | 1.34 (0.84, 2.13) | 1.22 (0.57, 2.64) | 1.12 (0.79, 1.61) | 0.99 (0.67, 1.45) | 0.50 (0.18, 1.37) | 1.22 (0.55, 2.73) | 0.75 (0.30, 1.88) | 1.38 (0.70, 2.75) |
| **Standard of care provided** | Not reduced (Ref) | 1.00 | N/a | 1.00 | N/a | 1.00 | N/a | 1.00 | N/a | 1.00 | N/a | 1.00 | N/a |
| | Reduced | **1.45 (1.20, 1.76)** | N/a | 0.97 (0.60, 1.56) | N/a | 0.99 (0.64, 1.52) | N/a | 1.14 (0.84, 1.54) | N/a | 0.70 (0.30, 1.65) | N/a | 0.92 (0.60, 1.41) | N/a |
| **Potentially morally injurious events** | No exposure (Ref) | 1.00 | 1.00 | 1.00 | 1.00 | 1.00 | 1.00 | 1.00 | 1.00 | 1.00 | 1.00 | 1.00 | 1.00 |
| | Exposure | **1.76 (1.43, 2.17)** | **1.54 (1.08, 2.20)** | 0.83 (0.57, 1.21) | 0.63 (0.38, 1.03) | 1.34 (0.98, 1.85) | 1.41 (0.82, 2.42) | 1.42 (0.80, 2.51) | 1.73 (0.97, 3.09) | 0.84 (0.33, 2.16) | 1.30 (0.60, 2.84) | 1.22 (0.73, 2.03) | 1.63 (0.84, 3.18) |

Statistically significant results are in bold; aOR: adjusted odds ratios–adjusted for age, sex, ethnicity, and date of survey completion (Time 2 models also adjusted for the corresponding outcome measured at Time 1); CI: confidence intervals

(PHQ-9 scores≥ 10). The sensitivity analysis revealed that at Time 1, for the clinical group, there remained evidence of an association between suicidal ideation and a lack of confidence about raising safety concerns and these concerns being addressed, as well as feeling unsupported by managers. At Time 2, evidence remained of an association between lack of confidence about safety concerns being addressed in the clinical group but not the non-clinical group.

In the cross-sectional analyses at Time 2 (S5 File), lack of access to PPE was associated with suicidal ideation, attempts and non-suicidal self-injury in the non-clinical groups. In this analysis feeling unsupported by managers and exposure to potentially morally injurious events were associated with increased suicidal ideation in both clinical and non-clinical groups.

## Discussion

In this study, we examined the evolution of suicidal thoughts and behaviour among a large sample of English healthcare workers (HCWs) during the pandemic and considered whether these outcomes were prospectively related to a range of occupational exposures. Five key findings emerged. First, during the first year of the COVID-19 pandemic, approximately one in ten participants reported having experienced suicidal thoughts in the previous two months. Over the same period, over 3% reported non-suicidal self-injury and over 2% reported having attempted suicide. However, due to a lack of pre-pandemic data, the extent to which these figures were influenced by the pandemic is unknown. Second, among HCWs who had not experienced suicidal thoughts at baseline, one-in-ten reported experiencing them six months later. Additionally, almost one-in-twenty-five HCWs reported having attempted suicide for the first time, whilst about one-in-sixteen HCWs reported first-time non-suicidal self-injury. Third, in the cross-sectional analyses, exposure to potentially morally injurious events, lack of confidence about raising safety concerns and the management of these concerns, feeling unsupported by managers, and providing a reduced standard of care were all associated with increased suicidal ideation in at least one of the surveys among clinicians and non-clinicians. Lack of access to PPE was associated with suicidal thoughts among non-clinical HCWs although not among clinical HCWs in both surveys. At Time 2, this association was also observed for suicide attempts and non-suicidal self-injury in non-clinical HCWs but not clinical HCWs. The differential associations between occupational groups may partially reflect residual confounding by factors such as socioeconomic status. Fourth, in the prospective analyses, a baseline lack of confidence about safety concerns being addressed predicted suicidal ideation at Time 2 among the clinical group. Fifth, contrary to our prior expectations [33, 34], re-deployment was not associated with suicidal thoughts or behaviour, but this may reflect the limited statistical power to test the association with redeployment status.

### Findings in the context of the wider literature

Differences in sample characteristics, the timeframe over which data were collected, and the specific survey questions preclude direct comparisons of STB prevalence estimates between studies. Nonetheless, our finding that approximately 30% of HCWs had ever previously experienced suicidal ideation, mirrors that of an online survey of 7,917 participants (92% from the UK, 50% HCWs) undertaken during the pandemic [35]. In this survey, 31% of HCWs reported previous suicidal ideation. Another study investigated suicidal thoughts among 709 HCWs working in Intensive Care units in England between June-July 2020. Suicidal thoughts that *"[they] would be better off dead, or of hurting [themselves] in some way several days or more frequently in the past 2 weeks"* were reported by 13% of HCWs [36].

Globally, there is considerable variation in prevalence estimates of STB in HCWs during the pandemic [12, 15]. In a rare follow-up study that provided data on first-time STB among 4,809 HCWs during the COVID-19 pandemic in Spain, the incidence risk of suicidal thoughts and behaviour combined was estimated to be 4% four months after initial data collection (May-September 2020 then October-December 2020) [15]. This study also investigated the relationship between risk factors and new cases of STB. We did not conduct similar analyses due to lack of statistical power.

Pre-pandemic data on suicidal behaviour among HCWs is mainly limited to suicide mortality rather than suicidal ideation or attempts [1]. However, a few European studies have investigated the latter outcomes among physicians [37–40]. The prevalence of lifetime suicidal ideation has ranged from 21.4% in Italy to 36.3% in Norway, whilst the prevalence of lifetime suicide attempts has ranged from 0% in Finland to 2.5% in Belgium. Between 11.1% (in

Norway) and 14.3% (in Italy) reported having experienced past-year suicidal ideation, and 0.3% of the Norwegian sample reported a suicide attempt in the previous year. The latter figure is considerably lower than the 3.9% incidence risk of first-time suicide attempts at Time 2 in this sample.

Among the general population in England, before the pandemic, 21% of people reported having experienced previous suicidal ideation [41]. Estimates of STB in the general population during the pandemic vary between surveys. One survey of the UK general population found that approximately one-in-ten people reported previous-week suicidal ideation during the first six weeks of the pandemic, and less than 1% reported previous-week suicide attempts [42]. In another UK general population survey, 18% reported suicidal ideation during the first month of the first national lockdown, whilst 5% reported self-harming during this period [43]. However, differences in the timing of data collection limit comparisons with the results in this study.

In keeping with previous systematic reviews [12, 13], our study identified associations between a range of potentially modifiable occupational factors and an increased risk of STB. Furthermore, exposure to potentially morally injurious events has been found to be associated with suicidal ideation in HCWs in the USA during the pandemic [44]. In a separate analysis of NHS CHECK data [20, 45], which investigated mental health outcomes, lack of support from managers and morally injurious events were both associated with psychological distress at multiple time periods throughout the pandemic. Non-availability of PPE was associated with psychological distress early in the pandemic [20].

In the UK, work-related suicides, unlike other work-related deaths, do not need to be formally reported and data on their occurrence are not systematically collected. A recent in-depth analysis qualitative analysis examined twelve suicide cases in the UK with a range of occupational backgrounds (including two doctors and one nurse) in which occupational factors had contributed [46]. Alongside identifying several key occupational factors, such as unmanageable workloads, the study highlighted the need for a consistent organisational response to work-related suicides.

## Strengths and limitations

This large-scale study provides an insight into the prevalence, incidence risk of STBs, and their relationship with occupational factors, among NHS HCWs in England during a time of unprecedented concern during the COVID-19 pandemic. The availability of six-month follow-up data has enabled estimation of the incidence risk of (first-time reported) STB. Other strengths of this study include: 1) the full sample was weighted based on the demographic characteristics of the total workforce in participating NHS Trusts, increasing the representativeness of the findings; 2) the analyses accounted for clustering of observations within the dataset; 3) the large sample size enabled stratification by clinical and non-clinical groups despite the investigation of relatively uncommon outcomes.

The relationship between depression, occupational factors, and suicidality is likely to be complex. The occurrence of depression may lie on the causal pathway between occupational factors and the development of suicidal thinking and behaviour. For this reason, in our primary analysis, we did not adjust for depression. Yet, conceivably, depression may also have confounded any association between occupational factors and suicidality i.e., people with depression may appraise occupational factors more negatively than people without, thus explaining any association between occupational factors and suicidality. For this reason, as a sensitivity check, we repeated the analyses adjusting for the effects of depression. Even after adjustment, key associations between occupational factors and suicidality remained.

However, there are several limitations to consider. First, the prevalence of mental distress is often higher in occupational cohorts than in population cohorts, perhaps due to an information bias whereby knowledge of participation in an occupational cohort increases reporting of symptoms [47]. Second, our understanding of changes over time in this study is limited by the following factors: 1) there was an overlap in timing between data collection for the two time periods; 2) pre-pandemic data were not collected; 3) the time period specified in the STB questions unintentionally decreased from the previous two months to the previous one month, which may have reduced the numbers who reported STB at Time 2; 4) some of the explanatory variables were only enquired about at one time period; 5) the number of HCWs almost halved between samples, and in the second time period there was also a large amount of missing data for two variables (concerns about the standard of care provided and access to PPE), resulting in imprecision in some estimates.

There are also limitations to consider regarding the associations between demographic characteristics and occupational factors and STB. We adjusted for only a few key confounders to avoid the possibility of overadjustment bias [48]. As a result, our findings may be affected by residual confounding.

Finally, the survey was completed by less than a fifth of eligible HCWs. This is higher than the overall response rate of 11.7% reported in the repeated survey that investigated the four-month incidence of STB among HCWs in Spain [15]. Other UK studies of HCWs conducted during this time have not reported a response rate [49]. Low response rates may reflect a variety of factors including: the pressures on the workforce at the time that the survey was conducted; HCWs' disinterest; stigma around mental health issues, and lack of awareness about the survey. The influence of STB on survey completion is unclear. For example, some people with STB may have been more likely to participate to share their experiences, whilst others may have been less likely due to being off work at the time of the survey or a lack of motivation. Nonetheless, we used weighted data to minimise response bias.

## Implications

There have been longstanding concerns about the risk of suicide among HCWs [1], although this risk varies by factors such as occupation and sex [50, 51]. While we identified a relationship between occupational factors and STB during the COVID-19 pandemic, many of the associations identified are likely to remain in operation even after the pandemic. The role of depression in relation to the association between occupational factors and suicidality is unclear, yet improvements in both mental health support and occupational factors in the workforce are very likely to be important. A number of such improvements have already been outlined elsewhere [52, 53]. Enhanced peer support provision and training managers to be able to communicate effectively about mental health are key modifiable factors [54]. Additionally, support for managers, themselves, should not be overlooked. Where occupational factors cannot be avoided (e.g., re-deployment), improved communication and reflective practice before or after the occurrence may be helpful.

Furthermore, the difference in findings between the clinical and non-clinical samples regarding personal protective equipment in our study highlights the importance of ensuring that the needs of non-clinical HCWs are adequately considered in occupational health and safety planning.

## Conclusion

Our findings suggest that occupational factors, such as exposure to potentially morally injurious events and the provision of a reduced standard of care, contribute to staff distress and

could increase the likelihood of suicidal thoughts developing. Improvements in mental health support and occupational exposures may reduce suicidal thoughts and behaviour among healthcare workers.

## Supporting information

**S1 File. Occupational groupings.**
(DOCX)

**S2 File. Survey questions relating to individual occupational factors.**
(DOCX)

**S3 File. Sensitivity analysis adjusting for PHQ9 in models (in Table 5) examining the association between baseline occupational factors and suicidal-related outcomes at Time 1 and Time 2.**
(DOCX)

**S4 File. Cross-sectional analyses at Time 2 of association between HCW demographic characteristics and suicidal ideation, suicide attempts, and non-suicidal self-injury, stratified by occupational role.**
(DOCX)

**S5 File. Cross-sectional analyses at Time 2 of association between HCW occupational factors and suicidal ideation, suicide attempts, and non-suicidal self-injury, stratified by occupational role.**
(DOCX)

**S6 File. Number and proportion of HCWs reporting exposure to occupational factors and probable depression at Time 1, stratified by occupational group.**
(DOCX)

## Acknowledgments

We are especially grateful to all the participants who took part in the study. We wish to acknowledge the National Institute of Health Research (NIHR) Applied Research Collaboration (ARC) National NHS and Social Care Workforce Group, with the following ARCs: East Midlands, East of England, South West Peninsula, South London, West, North West Coast, Yorkshire and Humber, and North East and North Cumbria. They enabled the set-up of the national network of participating hospital sites and aided the research team to recruit effectively during the COVID-19 pandemic.

The NHS CHECK consortium includes the following site leads: Siobhan Coleman, Sean Cross, Amy Dewar, Chris Dickens, Frances Farnworth, Adam Gordon, Charles Goss, Jessica Harvey, Nusrat Husain, Peter Jones, Damien Longson, Paul Moran, Jesus Perez, Mark Pietroni, Ian Smith, Tayyeb Tahir, Peter Trigwell, Jeremy Turner, Julian Walker, Scott Weich, Ashley Wilkie.

The NHS CHECK consortium includes the following co-investigators and collaborators: Peter Aitken, Ewan Carr, Anthony David, Mary Jane Doherty, Sarah Dorrington, Rosie Duncan, Sam Gnanapragasam, Cerisse Gunasinghe, Stephani Hatch, Danielle Lamb, Daniel Leightley, Ira Madan, Richard Morriss, Isabel McMullen, Dominic Murphy, Martin Parsons, Catherine Polling, Alexandra Pollitt, Anne-Marie Rafferty, Rebecca Rhead, Danai Serfioti, Chloe Simela, Charlotte Wilson Jones.

## Author Contributions

**Conceptualization:** Prianka Padmanathan, Danielle Lamb, Simon Wessely, Paul Moran.

**Formal analysis:** Prianka Padmanathan.

**Funding acquisition:** Sharon Stevelink, Neil Greenberg, Matthew Hotopf, Rosalind Raine, Simon Wessely.

**Methodology:** Prianka Padmanathan, Danielle Lamb, Paul Moran.

**Supervision:** Danielle Lamb, Paul Moran.

**Validation:** Danielle Lamb.

**Writing – original draft:** Prianka Padmanathan.

**Writing – review & editing:** Danielle Lamb, Hannah Scott, Sharon Stevelink, Neil Greenberg, Matthew Hotopf, Richard Morriss, Rosalind Raine, Anne Marie Rafferty, Ira Madan, Sarah Dorrington, Simon Wessely, Paul Moran.

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
