## [Decision Letter · Decision Letter 0]

7 Feb 2023

PONE-D-22-25700Suicidal thoughts and behaviour among healthcare workers in England during the COVID-19 pandemicPLOS ONE

Dear Dr. Padmanathan,

Thank you for submitting your manuscript to PLOS ONE. After careful consideration, we feel that it has merit but does not fully meet PLOS ONE’s publication criteria as it currently stands. Therefore, we invite you to submit a revised version of the manuscript that addresses the points raised during the review process.

We look forward to receiving your revised manuscript.

Kind regards,

Fentie Ambaw Getahun, PhD

Academic Editor

PLOS ONE

Journal Requirements:

MH, RR, and SW are senior NIHR Investigators.

SW has received speaker fees from Swiss Re for two webinars on the epidemiological impact of COVID 19 pandemic on mental health.

RR reports grants from DHSC/UKRI/ESRC COVID-19 Rapid Response Call, grants from Rosetrees Trust, grants from King's Together  rapid response call, grants from UCL (Wellcome Trust) rapid response call,  during the conduct of the study; & grants  from NIHR outside the submitted work.

PM reports grants from NIHR outside the submitted work.

MH reports grants from DHSC/UKRI/ESRC COVID-19 Rapid Response Call, grants from Rosetrees Trust, grants from King's Together  rapid response call, grants from UCL Partners rapid response call,  during the conduct of the study; grants from Innovative Medicines Initiative and EFPIA, RADAR-CNS consortium, grants from MRC, grants from NIHR, outside the submitted work.

SS reports grants from UKRI/ESRC/DHSC, grants from UCL, grants from UKRI/MRC/DHSC, grants from Rosetrees Trust, grants from King's Together Fund, and an NIHR Advanced Fellowship [ref: NIHR 300592] during the conduct of the study.

NG reports a potential COI with NHSEI,  during the conduct of the study;  he is the managing director of March on Stress Ltd which has provided training for a number of NHS organisations. NG is not aware if the company has delivered training to any of the participating trusts, as he is not involved in commissioning specific pieces of work.

DL is funded by the NIHR ARC North Thames. The views expressed in this publication are those of the authors and not necessarily those of the National Institute for Health Research or the Department of Health and Social Care.

The views expressed are those of the authors and not necessarily those of the NHS, the NIHR, or the Department of Health and Social Care.

Other authors report no competing interests.

Reviewers' comments:

Reviewer's Responses to Questions

**Comments to the Author**

1. Is the manuscript technically sound, and do the data support the conclusions?

Reviewer #1: Yes

Reviewer #2: Yes

2. Has the statistical analysis been performed appropriately and rigorously? 

Reviewer #1: Yes

Reviewer #2: No

3. Have the authors made all data underlying the findings in their manuscript fully available?

Reviewer #1: Yes

Reviewer #2: Yes

4. Is the manuscript presented in an intelligible fashion and written in standard English?

Reviewer #1: Yes

Reviewer #2: Yes

5. Review Comments to the Author

Reviewer #1: I think it is an interesting article which focus on vulnerable group of professionals who are working in covid-19 pandemic to assess suicidal behavior. it has an impact on professionals who are working in special occasions like COVID pandemic to take immediate actions and to alleviate suffering from this hidden killer behavior suicide. i recommended it to publish as it is

with regards!

Reviewer #2: I would like to thank you for giving me the chance to review this manuscript. The paper reports a large prospective study on incidence risk and prevalence of suicidal thoughts and behavior, and their relationship with demographic and occupational risk factors. The paper is well organized and clearly written. Nevertheless, there are some methodological issues that need to be addressed. My comments on the different sections of the paper are presented below.

Abstract

• May be useful to include the type of design used in the title

• The authors argued that there has been speculation regarding increase risk in suicide among healthcare workers during COVID-19. Are there no any preliminary empirical evidence on this?

• Is there any particular reason to stratify results by occupational role? I thought that this can be included in the regression model as a variable.

• It is not clear whether the design is longitudinal or repeated cross-sectional in the abstract

• I would suggest to highlight the baseline findings in the abstract

• I would suggest conclusion instead of discussion in the abstract.

• There is inconsistency between the results and conclusion sub-section s. While the results section highlights prevalence and incidence, the conclusion emphasized on factors associated with the outcome variables.

Background

• The authors claimed “Many studies from across the world have investigated the impact of the pandemic on suicidal thoughts and behavior.” However, they did not summarize what these studies found out.

• There is need to synthesize the findings of previous studies in the area (i. e. the prevalence of suicide thoughts and behaviors and contributing factors during the COVID-19 pandemic and other related pandemics). The authors need to justify this is worth studying.

• Overall, the background section is a bit shallow and doesn’t comprehensively synthesize the existing evidence and indicate the research gap.

Methods

• The authors started the methods section with describing the online survey. I would suggest this to come after describing about the design of the study, the study context and participants.

• Would 5% response rate be acceptable? I would say that those who have STB are likely to complete the survey more than those who have not and this is likely to overestimate the prevalence and incidence reported in this study.

• The psychometric properties and acceptability and feasibility of the three questions used to measures STB are not described. Did previous studies use these questions? Are they validated? What are the item response characteristics or qualities of these questions?

• The authors indicated that they used two-level random-effects logistic regression models. Is this a type of multilevel analysis? From the description of the analysis procedures, one can understand the need for a multilevel analysis (Trust is included in the model as explanatory variable and HCWs are clustered within each Trust).

Results

• There seems inconsistency in reporting the response rate: 5% in the methods section and 16% in the results section

• It is not clear what the authors are meant by the Time1 full cohort. It possible to understand from the data presented that only Time 1 sample and Time 2 sample responded to questions related to STB. I don’t think the Time 1 full cohort is related to any of the analyses done in this study.

• Is there any rationale for the age categories used in this study?

• I would suggest organizing the results section into subjects (such as prevalence, incidence risk, factors associated etc….). This will make easy for the reader to follow.

• It is not clear why separate models were estimated for the socio-demographic characteristics and occupational factors. For the same outcomes, estimating several models will increase the type error.

• It is necessary to make explicit (in the methods section) the fact that cross-sectional analyses were done for investigating the relationship between socio-demographic/occupational characteristics and STB.

Discussion

• The authors claimed that they did prospective relationship between socio-demographic and occupational factors and STB (Page 17, line 303). I don’t think this is correct. I didn’t see any prospective association analyses. Time 1 and Time 2 reports of STB are separately associated with socio-demographic and occupational factors.

• The findings of the study reported in this paper are compared with several previous studies conducted in different parts of the world. However, these studies are not synthesized and reported in the background section to give context to the problem and indicate the research gap.

6. PLOS authors have the option to publish the peer review history of their article (what does this mean?). If published, this will include your full peer review and any attached files.

Reviewer #1: No

Reviewer #2: **Yes: **Kassahun Habtamu

---

## [Decision Letter · Decision Letter 1]

11 May 2023

Suicidal thoughts and behaviour among healthcare workers in England during the COVID-19 pandemic: a longitudinal study

PONE-D-22-25700R1

Dear Dr. Padmanathan,

We’re pleased to inform you that your manuscript has been judged scientifically suitable for publication and will be formally accepted for publication once it meets all outstanding technical requirements.

Kind regards,

Charlotte Lennox

Academic Editor

PLOS ONE

Additional Editor Comments (optional):

Reviewers' comments:

Reviewer's Responses to Questions

**Comments to the Author**

1. If the authors have adequately addressed your comments raised in a previous round of review and you feel that this manuscript is now acceptable for publication, you may indicate that here to bypass the “Comments to the Author” section, enter your conflict of interest statement in the “Confidential to Editor” section, and submit your "Accept" recommendation.

Reviewer #1: All comments have been addressed

Reviewer #2: All comments have been addressed

2. Is the manuscript technically sound, and do the data support the conclusions?

Reviewer #1: Yes

Reviewer #2: (No Response)

3. Has the statistical analysis been performed appropriately and rigorously? 

Reviewer #1: Yes

Reviewer #2: (No Response)

4. Have the authors made all data underlying the findings in their manuscript fully available?

Reviewer #1: Yes

Reviewer #2: (No Response)

5. Is the manuscript presented in an intelligible fashion and written in standard English?

Reviewer #1: Yes

Reviewer #2: (No Response)

6. Review Comments to the Author

Reviewer #1: I am satisfied with topic Suicidal thoughts and behaviour among healthcare workers in England during the

COVID-19 pandemic: a longitudinal study the statstical analysis and intepreation as well

Reviewer #2: (No Response)

7. PLOS authors have the option to publish the peer review history of their article (what does this mean?). If published, this will include your full peer review and any attached files.

Reviewer #1: No

Reviewer #2: No

---

## [Editor Report · Acceptance letter]

31 May 2023

PONE-D-22-25700R1 

Suicidal thoughts and behaviour among healthcare workers in England during the COVID-19 pandemic: a longitudinal study 

Dear Dr. Padmanathan:

I'm pleased to inform you that your manuscript has been deemed suitable for publication in PLOS ONE. Congratulations! Your manuscript is now with our production department. 

Kind regards, 

on behalf of

Dr. Charlotte Lennox 

Academic Editor

PLOS ONE